# Attending a Sports Club Can Help Prevent Visual Impairment Caused by Cram School in Elementary School Children in Japan

**DOI:** 10.3390/ijerph182312440

**Published:** 2021-11-26

**Authors:** Yui Mineshita, Hyeon-Ki Kim, Takae Shinto, Mai Kuwahara, Shigenobu Shibata

**Affiliations:** 1Graduate School of Advanced Science and Engineering, Waseda University, 2-2 Wakamatsu-Cho, Shinjuku-Ku, Tokyo 1628480, Japan; m-yk_1426@fuji.waseda.jp (Y.M.); hk.kim@aoni.waseda.jp (H.-K.K.); y53-m-h423@moegi.waseda.jp (T.S.); kmykmya@akane.waseda.jp (M.K.); 2Laboratory of Physiology and Pharmacology, Faculty of Science and Engineering, Waseda University, 2-2 Wakamatsu-Cho, Shinjuku-Ku, Tokyo 1628480, Japan

**Keywords:** glasses, elementary school students, sports club, cram school

## Abstract

Longer durations for near-work activities, such as studying, worsen eyesight. In contrast, outdoor exercise is effective in reducing the risk of developing myopia. Despite these findings, however, the interaction between studying and exercise in eyesight has not been quantitatively evaluated. Moreover, since there is a culture of attending lessons in Japan, it is important to investigate the relationship between elementary school activities, such as cram schools or sports clubs, and vision. Therefore, in this study, we examined whether attending cram schools and/or sports clubs is associated with the use of glasses among elementary school students. We conducted a survey among 7419 elementary school students in Tokyo, Japan using a food education questionnaire. A logistic regression analysis was used to evaluate the relationship between wearing glasses, an objective variable, and attending sports clubs and cram schools. Sex and school year were considered confounding factors. The results of this study showed that students who attended only sports clubs were more likely to be categorized into the “not wearing glasses” group (*p* = 0.03, OR = 1.45), whereas those who attended only cram schools were more likely to be categorized into the “wearing glasses” group (*p* = 0.008, OR = 0.67). In addition, students who attended both cram schools and sports clubs were more likely to be categorized into the “not wearing glasses” group than those who only attended cram schools (*p* = 0.28, OR = 0.85). Our findings indicate that attending not only cram schools but also sports clubs may prevent deterioration of eyesight. Parents and health care providers need to take these findings into account in order to prevent visual impairment in children.

## 1. Introduction

Myopia, the leading cause of visual impairment, is estimated to affect half of the world’s population by 2050 [1,2]. In particular, the prevalent rates of myopia in East and Southeast Asia are the highest worldwide [3,4,5,6]. In parallel with the increase in overall myopia, there has also been an increase in the prevalence of intense myopia, which is related to increased visual impairment and blindness, mainly due to retinal chorioretinal degeneration and retinal detachment [7,8]. In Japan, the percentage of elementary school students with visual acuity of <1.0% has been increasing every year [9]. Moreover, the economic burden of uncorrected distance refractive errors, mainly caused by myopia, has been estimated to be approximately USD 202 billion per year, making it a significant economic burden [10]. In general, the incidence of myopia increases during school-age years and decreases in later adulthood [11]; therefore, it is important to prevent myopia in school-age children.

Genetic factors are thought to be involved in the development of visual disorders such as myopia, but the rapidly increasing prevalence of myopia has been attributed to environmental and lifestyle factors [1]. Previous studies have shown an association between visual impairment and risk factors, such as near-work activities, including studying and reading [12]. Furthermore, near work has been proposed as a factor in the progression of myopia, and it has been reported that near work, such as intense computer work, increases visual impairment [13,14]. In contrast, outdoor activities reduce the risk of developing visual impairment; many studies have investigated the relationship between outdoor activities and myopia [15,16,17,18,19]. A previous study reported the association between sports and outdoor activities and myopia, and showed that increased time spent in sports and outdoor activities is associated with decreased myopia [20]. In addition, both cross-sectional and longitudinal studies have reported that participation in sports and outdoor activities has a preventive effect on myopia [21,22]. However, to the best of our knowledge, no previous studies have quantitatively evaluated the interaction between studying and exercise.

In Japan, there is a culture of attending lessons at an early age, and according to the 2007 report of the Ministry of Education, Culture, Sports, Science, and Technology, >80% of elementary school students have taken some kind of lessons [23]. Moreover, there are many elementary school students in Japan who are preparing for junior high school entrance exams, with >45% of them engaging in learning activities outside of school, including cram schools, correspondence correction, and private tutoring [23]. According to a previous study, more elementary school students in Minato city (“Minato-ku” in Japanese), where this survey was conducted, tend to take junior high school entrance exams than those in other wards in Tokyo [24]. Furthermore, approximately 60% of elementary school students attend sports clubs [23]. Therefore, it is important to investigate the relationship between attending cram schools and/or sports clubs and vision among elementary school students.

In the present study, we investigated whether attending cram schools and/or sports clubs is associated with the use of glasses among elementary school students. This study is expected to discover not only the current state of the use of glasses in Japanese children but also the relationship between attending cram schools and sports clubs and the state of the use of glasses. We focused on cram school and sports club activities to test the hypothesis that cram school and sports club attendance may be associated with the state of the use of glasses, given the high rate of cram school and sports club attendance among Japanese children.

## 2. Materials and Methods

### 2.1. Participants and the Super Diet Education (Shokuiku) Project

The Super Dietary Education Project, a dietary education project of the Japanese Ministry of Education, Culture, Sports, Science, and Technology (MEXT), aims to develop programs to promote dietary education among children in cooperation with various outside organizations. The overall objective of the project is to contribute to healthy living and improve the health of school children through nutrition education. In collaboration with this project, we conducted a cross-sectional study on a cohort of elementary school students using a survey questionnaire. The participants were 7419 children 6 to 12 years of age who belong to 18 public elementary schools in Minato City, Tokyo, Japan. This survey was conducted from 2018 to 2019. Cases were excluded from all analyses if missing any of the answers. This study’s protocol conformed to the Helsinki Declaration and was approved by the Ethics Review Committee on Research with Human Subjects of Waseda University, Tokyo, Japan (application no. 2019-195).

### 2.2. Questionnaire

For this survey, we used a questionnaire on food education (shokuiku) developed by the Minato City Board of Education. In Japanese, “shoku” means diet, and “iku” means growth and education. In total, 64 questions composed the questionnaire regarding basic attributes (sex, grade, etc.), learning ability, physical activity, lifestyle, eating habits, and food culture. Among the items in the questionnaire, information such as sex, grade, lessons, and whether the students were wearing glasses was collected and analyzed. Teachers at the schools explained the purpose of the study and distributed the questionnaires, which were completed by the children and their parents before returning them to the schools. Table 1 presents a summary of the questions and response options.

### 2.3. Sports Clubs and Cram Schools

Regarding the attendance in lessons outside of school, we specifically asked the participants, “What type of lessons you are attending?” and their answers were categorized from 1 to 5 (1, sports club; 2, cultural school; 3, cram school; 4, other; and 5, no lessons). Multiple answers were allowed for this question. However, since we only used data from elementary school students who answered 1, 3, both 1 and 3, or 5, the answers were divided into the following four groups: the “sports club” group (those who answered 1), the “cram school” group (those who answered 3), the “sports club and cram school” group (those who answered 1 and 3), and the “no lessons” group (those who answered 5). Participants reporting other answers (3897 (52.5%) children) were excluded from this analysis; thus, only data from 3522 children were analyzed. Furthermore, participants were queried regarding how many times a week they went to these lessons and how long they attend them (<1, 1–2, 2–3, ≥3 h). These questions were then used in the Fisher’s exact test separate from the logistic regression analysis.

### 2.4. Use of Glasses

Regarding the use of glasses, we specifically asked the participants, “Do you wear glasses?” and their responses were categorized as either 1 (yes) or 2 (no). Based on the responses, participants were divided into the “wearing glasses” group (those who answered 1) and the “not wearing glasses” group (those who answered 2).

### 2.5. Statistical Analyses

A data analysis was performed using the Predictive Analytics Software for Windows (SPSS Japan Inc., Tokyo, Japan). Calculating the total sample size for detecting a medium effect (f2 [effect size] = 0.15), a minimum sample size of 146 was required to achieve approximately 95% power to detect a large effect at a significance level of 0.05 (G*Power, version 3.1.9.2, Universitat Kiel, Kiel, Germany). The odds ratios (ORs) and 95% confidence intervals (CIs) were calculated. In this study, the chi-square test was used to compare sex and school year, which were used as confounding factors, between the groups. A logistic regression analysis was then performed to examine the results; two analyses were performed, with “wearing glasses” as the objective variable and “sports club” and “cram school” as the explanatory variables. First, we examined whether each explanatory variable was related to the objective variable using univariate models. Then, we performed multivariate logistic regression analyses for all variables showing a significant difference in the univariate models. Further statistical analysis was performed using Fisher’s exact test to analyze the relationship between wearing glasses and the number of sports clubs and cram schools attended and the time spent attending them. Results with *p*-values of less than 0.05 were considered significant.

## 3. Results

The characteristics of the participants are presented in Table 2. The analysis of the data collected from questionnaires completely answered by 3522 (47.47%) school children showed that there were significant differences in sex and school year between the “wearing glasses” and “not wearing glasses” groups.

Results of the multivariate analysis are presented in Table 3. The “sports club” group (OR = 1.45, 95% CI = 1.03–2.04) was positively correlated with wearing glasses, whereas the “cram school” group (OR = 0.67, 95% CI = 0.49–0.90) was negatively correlated with wearing glasses. In contrast, the “sports club and cram school” group was not correlated with wearing glasses. However, the OR of the “sports club and cram school” group was greater than that of the “cram school” group, showing that participants who attend sports clubs were more likely to be categorized into the “not wearing glasses” group. In contrast, participants who only attended cram schools were more likely to be categorized into the “wearing glasses” group, while those who attended both sports clubs and cram schools were more likely to be categorized into the “not wearing glasses” group.

In Table 4, the data of 3495 children were used because the elementary school students (*n* = 27) who did not answer sports club and cram school were excluded from the analysis. Table 4 summarizes the relationship between attending sports clubs and/or cram schools and rate of wearing glasses (Table 4), showing that the rate of wearing glasses increased with a higher cram school attendance. However, by attending sports clubs, the rate of wearing glasses was lower even with a higher cram school attendance. Participants who attended cram school once or twice a week had a low rate of wearing glasses, regardless of the number of times they attended sports clubs; thus, it was not possible to show a decrease in the rate of wearing glasses due to the effects of exercise. Similarly, participants who attended cram schools six or seven times a week cannot afford to attend sports clubs; thus, it was not possible to show a decrease in the rate of wearing glasses. In contrast to the previous two, for those who attended cram school three to five times a week, the rate of glasses use was found to be lower for those who also attended their sports club, indicating that the rate of glasses use decreased due to the effect of exercise.

In Table 5, the data of 3455 children were used because the elementary school students (*n* = 67) who did not answer the time of the sports club and the cram school were excluded from the analysis. We also summarized the relationship between the time spent attending one sports club and/or cram school session and the rate of wearing glasses (Table 5), showing that the rate of wearing glasses increased with a longer time spent attending one cram school session. However, neither an increase or decrease was confirmed regarding the time spent attending one sports club session; thus, the relationship between the time spent attending one sports club session and the rate of wearing glasses was not clarified. However, participants who study for >2 h in one cram school session and exercise for >1 h in one sports club session had a reduced rate of wearing glasses.

## 4. Discussion

### 4.1. Main Results

The present study surveyed elementary school children from Minato city to examine the relationship between wearing glasses and attending sports clubs and cram schools. The relationship between the three was found to be statistically significant. Students who only attended cram schools were more likely to wear glasses, whereas those who attended sports clubs were less likely to wear glasses. Further, attending both cram schools and sports clubs may prevent eyesight deterioration.

### 4.2. Relationship between Wearing Glasses and Attending Sports Clubs and Cram Schools

Previous studies have reported a positive correlation between the onset of myopia and study-time length [25]. Myopia is associated with risk factors such as near-work activities, which include studying and reading [12]. Proximal tasks such as close reading distance may contribute to the onset and progression of myopia as it may affect excessive lens convergence and growth of the ocular axial diameter [26]. In addition, exposure to display screen devices has increased in recent decades, especially with the massive use of computers and tablets for study and work. This near-universal and widespread exposure is thought to be a strong risk factor for the onset and progression of myopia and contributes to the increasing prevalence of myopia [13]. In this study, children who attended cram schools may have a long study time and more time to work in a close range. Therefore, it is reasonable that students who only attended cram schools and were exposed to such near-work activities were more likely to wear glasses.

In contrast, attending sports clubs instead of cram schools may prevent eyesight deterioration. Myopia is the most common cause of eye disease and visual impairment, and its prevalence has increased significantly in recent decades [1,27,28]. Therefore, it is very important to understand its pathogenesis and to identify potential interventions. Many studies on the relationship between outdoor activities and myopia have reported that outdoor activities were found to delay the onset of myopia. Sunlight was found to be deeply involved in this process as it stimulates the retina to release dopamine, consequently delaying the onset of myopia [29]. However, it should be noted the type of sports club was not specified in this study, and it was thought that the exercises performed included both outdoor and indoor activities. Previous studies have reported that although outdoor activity prevented the development of myopia, there was no association between daytime light intensity or time, timing, and frequency spent outdoors [30]. In addition, mouse experiments have reported that exercise reduces harmful ocular blood vessel overgrowth by up to 45% [31]. Therefore, it was considered that exercise itself may be effective in preventing the onset of myopia. Therefore, attending sports clubs, regardless of exercising indoors or outdoors, may prevent eyesight deterioration. However, it is necessary to collect detailed data, such as the type of sports and the location where the activity was performed, and reconsider the effect differences between indoor and outdoor exercises in future studies.

Although the mechanism is unclear, the frequency of sports club activity rather than its duration may be involved in the protection of eyesight deterioration induced by attending cram schools. Therefore, we believe that children would be better protected from myopia if they attend sports clubs with more frequency rather than exercising for a long duration. On the other hand, several previous studies showed that genetic factors play an important role in the development of myopia [32]. However, some studies have reported that heredity is only involved in a small percentage of myopia cases, and most agree that myopia is caused by both genetic and environmental factors [26,33]. The previous study showed that low to moderate myopia is mainly caused by environmental factors [33]. The state of glasses used in this study is current and does not reflect parental eyeglass wearing rates or the state of glasses used before entering elementary school. Therefore, a more detailed study that includes these factors is needed in the future.

In recent years, the myopia-inhibiting effect of violet light (360 nm to 400 nm, the short wavelength region of visible light), which is abundant in outdoor ambient light, has become clear, and it has been speculated that the violet light deficiency caused by the decrease in outdoor activity is one of the factors contributing to the rapid increase in myopia prevalence [34,35]. In fact, in a previous study, visible violet light was found to be effective in preventing the onset of myopia in humans [35]. Furthermore, the molecular mechanism of this effect is being elucidated, and it was clarified that violet light is received by OPN5 (neuropsin), a photoreceptor expressed in retinal ganglion cells in the inner layer of the mouse retina that is involved in local retinal circadian rhythms, intraocular vascular development, and regulation of deep body temperature, thereby suppressing myopia progression by maintaining choroidal thickness [36]. Therefore, the time of day of exercise may be important for the prevention of visual impairment.

Furthermore, a preliminary analysis showed that the rate of wearing glasses was low in participants attending cultural schools (17.6% for those who have not learned anything, and 17.3% for those who attend only cultural school), meaning that the relationship between cram schools and club schools should be considered important, since there was no effect of preventing myopia due to studying in these schools.

### 4.3. Limitations

Despite these results, there are several limitations to this study. First, the survey did not reflect the actual lifestyle and anthropometric data of the participants. Children may exaggerate or downplay aspects of their lifestyle based on social expectations. Second, this study relied on children’s responses, which may introduce errors due to differences in children’s interpretations of the questions. Third, since this study was conducted on elementary school children, it may not be applicable to middle school, high school, and university students or adults. Therefore, the results of this study may not be generalizable. Thus, future studies should expand the scope of this study. Fourth, this study was conducted in Minato city, Tokyo, Japan, and it is unclear whether the same results would be obtained in other areas. Therefore, the target area must be expanded in future studies as well. Fifth, the type, location, and time of the sports club activities were not included in the questionnaire, and thus, a detailed examination was not possible. In addition, since there are no data on how many years the cram school or sports clubs have been held, the effect of the habituation of cram schools are unknown. Finally, the present study examined the interaction between attending cram schools and/or sports clubs and the use of eyeglasses but did not clarify the causal relationship. Therefore, it is necessary to examine in detail the causal relationship between cram schools and sports clubs and the use of eyeglass (visual disorders) through cohort studies in the future.

## 5. Conclusions

This study provided data on the relationship between the state of glasses use and cram school and sports club activities among Japanese elementary school children. Our findings indicate that attending not only cram schools but also sports clubs may prevent the deterioration of eyesight. Therefore, parents, health care providers, and policymakers should pay more attention to these in order to prevent visual impairment in children and correct spectacle situations.

## Figures and Tables

**Table 1 ijerph-18-12440-t001:** Question content and options for the explanatory variable and objective variable.

Explanatory Variable	Contents of Question
Type of lessons	What type of lesson you are attending?
	1. sports club; 2. cultural school; 3. cram school; 4. other; 5. no lessons
**Objective Variable**	
Glasses	Do you wear glasses?
	1. yes; 2. no

**Table 2 ijerph-18-12440-t002:** Characteristics of elementary school students.

		Do You Wear Glasses?					
Item			Wearing Glasses		Not Wearing Glasses		
Age, Mean (SE)			10.4	(0.056)	9.01	(0.033)	
		*N*	*n*	%	*n*	%	*p*-Value ^a^
Sex	Male	2107	412	19.6	1695	80.4	0.013 *
	Female	1415	326	23.0	1089	77.0	
School grade	1st	613	37	6.0	576	94.0	<0.001 ***
	2nd	577	56	9.7	521	90.3	
	3rd	560	63	11.3	497	88.8	
	4th	574	126	22.0	448	78.0	
	5th	611	198	32.4	413	67.6	
	6th	587	258	44.0	329	56.0	

*** *p* < 0.001, * *p* < 0.05, ^a^ chi-square test.

**Table 3 ijerph-18-12440-t003:** Results of logistic regression analysis.

Types of Lessons	OR ^a^ (95% CI)
No lessons	1
Sports club	1.45 * (1.03–2.04)
Cram school	0.67 ** (0.49–0.90)
Sports club and cram school	0.85 (0.64–1.14)

** *p* < 0.01, * *p* < 0.05. ^a^ OR: odds ratio, 95% CI: 95% confidence interval.

**Table 4 ijerph-18-12440-t004:** Relationship between sports club and/or cram school attendance and the rate of wearing glasses.

Sports Club/ Cram School	0 Times		1 Times		2 Times		3 Times		4 Times		5 Times		6 Times		7 Times	
	*N*	%	*N*	%	*N*	%	*N*	%	*N*	%	*N*	%	*N*	%	*N*	%
0 times	466	17.6	85	11.8	207	16.4	199	39.2 ***	141	53.2 ***	81	49.4 ***	40	50.0 ***	32	53.1 ***
1 time	269	11.5 *	141	15.6	232	20.3	134	26.9 ^&^	52	44.2	19	31.6	4	50.0	3	66.7
2 times	214	15.0	102	11.8	177	18.6	99	20.2 ^&&^	32	18.8 ^###^	11	9.1 ^$^	3	66.7	4	25.0
3 times	125	9.6 *	81	11.1	93	16.1	46	26.1	9	0.00 ^##^	5	0.00	0	0.00	0	0.00
4 times	82	11.0	41	17.1	55	14.5	29	24.1	6	16.7	1	0.00	1	0.00	0	0.00
5 times	35	11.4	18	27.8	32	6.30	7	0.00 ^&^	2	0.00	1	100	0	0.00	0	0.00
6 times	27	7.40	11	36.4	11	0.00	1	0.00	1	0.00	2	50.0	0	0.00	0	0.00
7 times	13	0.00	3	0.00	8	0.00	2	0.00	0	0.00	0	0.00	0	0.00	0	0.00

* *p* < 0.05 vs. SC 0/CS 0, *** *p* < 0.001 vs. SC 0/CS 0, **^&^**
*p* < 0.05 vs. SC 0/CS 3, **^&&^**
*p* < 0.01 vs. SC 0/CS 3, ^##^
*p* < 0.01 vs. SC 0/CS 4, ^###^
*p* < 0.001 vs. SC 0/CS 4, ^$^
*p* < 0.05 vs. SC 0/CS 5. Fisher’s exact test was used for all statistics.

**Table 5 ijerph-18-12440-t005:** Relationship between the time spent attending one sports club and/or cram school session and the rate of wearing glasses.

Sports Club /Cram School	0 h		1 h		12 h		23 h		≥3 h	
	*N*	%	*N*	%	*N*	%	*N*	%	*N*	%
0 h	466	17.6	66	15.2	235	15.3	181	40.9 ***	294	52.0 ***
1 h	64	4.7 **	33	15.2	50	24.0	23	30.4	17	52.9
12 h	507	12.4 *	157	10.8	560	13.8	184	24.5 ^&&^	108	35.2 ^##^
23 h	129	13.2	24	20.8	102	13.7	55	21.8 ^&^	31	45.2
≥3 h	52	9.60	12	25.0	50	14.0	30	30.0	25	28.0 ^#^

* *p* < 0.05 vs. SC 0 h/CS 0 h, ** *p* < 0.01 vs. SC 0 h/CS 0 h, *** *p* < 0.001 vs. SC 0 h/CS 0 h, ^&^
*p* < 0.05 vs. SC 0 h/CS 2–3 h, ^&&^
*p* < 0.01 vs. SC 0 h/CS 2–3 h, ^#^
*p* < 0.05 vs. SC 0 h/CS SC 0 h/CS 35 h, ^##^
*p* < 0.01 vs. SC 0 h/CS ≥ 3 h. Fisher’s exact test was used for all statistics.

## Data Availability

The datasets for the current study are available from the corresponding author upon reasonable request.

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
