# Peer review of "Attending a Sports Club Can Help Prevent Visual Impairment Caused by Cram School in Elementary School Children in Japan"

_ijerph, 2021, doi:10.3390/ijerph182312440_

Round 1
Reviewer 1 Report
Title
I like the title of the paper. The title might sport interest in the reader and is accurate presentation of the paper.
Abstract
Overall, abstract provides all important information for the reader in short. However, I suggest Authors to add some specific results to the abstract (with specific values). Also, Authors could add one sentence about previous research (at the beginning) and one sentence about future direction (at the end).
Introduction
Overall, the introduction provides a great overview of the current literature. On the other hand, the introduction is clearly too short, and I suggest Authors to find more literature on this topic. Also, I suggest Authors to provide section “the present study” in which Authors could present the purpose of the present study (currently it is only one sentence – it is clearly too short). The manuscript could also benefit from adding specific hypotheses for the study.
Materials and Methods
Please add the standard deviation of participants’ age.
Please provide more information about the questionnaire used in this study. It is not enough to say that “The questionnaire used in this experiment was the same as that used in our previous study”. Is this questionnaire a valid and reliable instrument?
Please revise the part in which you mention the software which was used. Currently it is not fully adequate.
Table 2 – presentation of school year, and boys and girls are not adequate, please revise. Also, please fit this table on one page.
Table 3 – please revise table based on the guidelines, it is not currently adequate.
Table 4 – please fit this table on one page.
Table 5 – I see blue and red coloured text in this table, please revise.
Discussion
Overall, the discussion is good, Authors discuss the results from several angels and put it into proper context. But again, the discussion is too short, please revise.
Conclusions section is too short, please revise.
Supplementary Materials has yellow background, please revise.
Reviewer 2 Report
The article “Attending sports clubs attenuate the incidence of visual impairment in elementary school children in japan” is a particular research article where the authors analyse the relation between wearing or not glasses depending on attending cram school or sports clubs.
The article is almost clear and well-written, even though some revisions are desirable.
- In line 91, the number of excluded children is 2926 with a percentage of 39.4%. The percentage does not coincide with the total number of subjects analysed in this study, does it?
- At line 163 should be better to reformulate the entire sentence; “Had a reduction of the rate of wearing glasses” or “had a reduced rate of wearing glasses”.
- The data presented in Tables 4 and 5 (row 151 and 164 respectively) do not correspond to the total number of subjects analysed in this study. That is, in Table 4 the total number of subjects analysed is 3495 instead of 3522, and in Table 5 the total number of subjects analysed is 3455 instead of 3522. Why were the remaining subjects excluded?
- In addition, the interpretation of Tables 4 and 5 and of the division between CS and SC is not clear.
- In the description of Table 1 on line 169 there is a printing error of "cram school".
- The conclusions could be implemented and are different from those in abstract.
- Inconsistent bibliography in relation to the guidelines of article.
- In the “limitations” paragraph too many limitations are stated, so the title could be less incisive and report a sentence as written in the “conclusions”.
- In line 222 the authors should cite a study on dawn light in humans.
- In the “discussion” authors report that “students who only attended cram schools were more likely to wear glasses, whereas those who attended sports clubs were less likely to wear glasses. Further attending both cram schools and sports club may prevent eyesight deterioration”. This statement seems to be a too forced conclusion as it does not take into account visual problems prior to the beginning of primary school.
- It ignores the fact that children may or may not play sport influenced by wearing glasses; the uncomfortableness of wearing them may limit or discourage them from playing sport. some social factors had to be taken into consideration before doing the study.
- - In addition, the questionnaire retrieved from the previous work lacked questions on the use of glasses and the club attended (questions present in this study).
Reviewer 3 Report
Although the study involved data collection from a large number of people, the main question is about the association between the use of glasses and the practice cram schools and/or to sports clubs. I can't see an association between the two variables. I believe the main point of the study would be to verify whether people who attend classes wear glasses or not. However, there is no cause and consequence relationship.
Round 2
Reviewer 1 Report
Authors have done well job by addressing all the comments raised by the Reviewers.
Author Response
Thank you for your very careful review of our paper, and for the comments, corrections, and suggestions that ensued.

Reviewer 3 Report
The authors were not able to answer my question about the rationale of the study, suggesting future studies.
